# Development of a Non-Peptide Angiotensin II Type 1 Receptor Ligand by Structural Modification of Olmesartan as a Biased Agonist

**DOI:** 10.3390/biomedicines11051486

**Published:** 2023-05-19

**Authors:** Yoshino Matsuo, Yasunori Suematsu, Hidetaka Morita, Shin-ichiro Miura

**Affiliations:** 1Department of Cardiology, Fukuoka University School of Medicine, Fukuoka 814-0180, Japan; 2Department of Internal Medicine, Fukuoka University Nishijin Hospital, Fukuoka 814-8522, Japan

**Keywords:** AT_1_ receptor, biased ligand, inositol phosphate production, extracellular signal-regulated kinase

## Abstract

As a biased agonist, peptide angiotensin II (Ang II) type 1 (AT_1_) receptor ligand antagonizes Ang II-stimulated G protein signaling but stimulates several kinase pathways. Here, we developed a non-peptide AT_1_ receptor compound as a biased ligand. We synthesized three non-peptide AT_1_ receptor ligands (R239470, R781253, and R794847) as candidates of biased ligands. Extracellular signal-regulated kinase (ERK) 1/2 activation and inositol phosphate (IP) production were measured using a cell system that overexpressed AT_1_ receptors (wild-type, L112A, Q257A, Y292A, and N295A receptors). We also examined the modes of receptor–ligand binding using a competition binding study. The *K*_d_ values of R239470, R781253, and R794847 for the AT_1_ wild-type receptor were 0.8, 21, and 48 nM, respectively, as assessed in a competition binding study. Those of R239470, R781253, and R794847 for the L112A receptor were 37, 23, and 31 nM, respectively. R781253 and R794847 decreased and increased IP production, respectively, whereas R239470 did not change IP production. R781253 and R794847, but not R239470, activated ERK1/2. In conclusion, R239470, R781253, and R794847 act as a neutral antagonist, an inverse agonist, and an agonist with regard to IP production, respectively. On the other hand, R781253 and R794847, but not R239470, are agonists toward ERK1/2 activation. Thus, we developed a non-peptide AT_1_ receptor compound as a biased ligand.

## 1. Introduction

Angiotensin II (Ang II) type 1 (AT_1_) receptor, which is a member of the G-protein-coupled receptor (GPCR) family, mediates most known pathological cardiovascular functions in addition to physiological functions [1,2]. There are many interesting aspects of the functions of GPCRs [3], such as homo- and hetero-oligomerization, constitutive or spontaneous activity in the absence of agonists, nuclear localization, trafficking, intracellular protein-evoked receptor activation, and the GPCR receptor-associated proteins. Studies on these topics may lead to new and improved therapeutics for cardiovascular diseases.

AT_1_ receptor blockers (ARBs) block the diverse effects of Ang II, such as vasoconstriction and cardiac hypertrophy. A biased ligand is a ligand that, when bound, activates the β-arrestin signal but not the Gaq signal, for example [4,5]. We previously reported that [Sar^1^, Ile^4^, Ile^8^]Ang II, which is an Ang II analogue, did not increase Gq-dependent inositol phosphate (IP) production. On the other hand, it still activated Gq-independent extracellular signal-regulated kinase (ERK) 1/2 in human coronary artery smooth muscle cells, as well as in a stably expressed AT_1_ receptor cell line [5]. Several ARBs show variable efficacies in addition to anti-hypertensive effects toward protection against organ damage in diabetic nephropathy, cardiac hypertrophy, arrhythmia, and renal failure [6]. Over the past several years, the efficacies of ARBs have been compared and differences have been observed, such as in reducing left ventricular mass, increasing nitric oxide production, lowering monocyte chemoattractant protein-1 and plasma plasminogen activator inhibitor type-1 antigen, and inhibiting platelet activation, except with regard to the lowering of blood pressure (as a pleiotropic effect) [7]. In addition, many studies have suggested that additional tissue-protective benefits of ARB therapy may be mediated by AT_1_ receptor β-arrestin signaling, although the structural basis for the differential activation of this mechanism by ARBs is not known [8,9].

Peptide-type AT_1_ receptor-selective agonists, which are biased ligands, have been developed, and one of them, TRV120027 (Sar-Arg-Val-Tyr-Ile-His-Pro-DAla-OH), competitively antagonizes Ang-II-stimulated G protein signaling. It stimulates β-arrestin recruitment and activates several kinase pathways including ERK 1/2, Src, and so on [10]. In addition, the AT_1_ receptor biased ligands TRV120023 and TRV120067 were reported to be useful in a mouse model of familial dilated cardiomyopathy [11] and in a mouse model of acute heart failure and dilated cardiomyopathy [12], respectively. On the other hand, a phase II trial with TRV027, which is similar to TRV120067, did not show an improved prognosis at day 30 compared with a placebo in patients with heart failure [13]. Peptide forms may be readily degradable, unstable, and inconsistent in their efficacy. In addition, although it activates the β-arrestin pathway, which is useful for treating heart failure, it is not clear whether it can activate other pathways that exacerbate heart failure because of the diversity of intracellular signals. If non-peptide AT_1_ receptor ligands could be developed, it would be possible to verify these points in detail. Most ARBs have common molecular structures (biphenyl-tetrazol and imidazole groups), and it is clear that ARBs have ‘class effects’ [14]. On the other hand, recent studies have demonstrated that not all ARBs have the same effects. Some benefits conferred by ARBs may not be class effects, and instead may be ‘molecular effects’ [14]. In addition, small differences in the chemical structures of ligands can be responsible for agonism, neutral antagonism, or inverse agonism toward a GPCR. A small difference in the molecular structure of angiotensin II receptor blockers induces agonism, neutral antagonism, or inverse agonism toward AT_1_ receptor [15]. Based on these reports, we thought that it may be possible to develop non-peptide AT_1_ receptor biased ligands by slightly changing the structure of ARBs.

Therefore, by using olmesartan as the basic structure of ARBs, we created compounds with various slight structural changes and used them as candidates of biased ligands. For example, the carboxyl group of olmesartan was converted to a carbamoyl group, which eliminated its potent inverse agonism. Instead, we designed and produced an olmesartan-related compound by adding a 4-hydroxybenzyl group to the biphenyl group so that it may have agonistic action. Here, we developed a non-peptide AT_1_ receptor compound as a biased ligand.

## 2. Materials and Methods

### 2.1. Materials

We purchased the following reagents: Ang II and [Sar^1^, Ile^8^]Ang II (Peptide Institute Inc., Osaka, Japan); olmesartan (Toronto Research Inc., Toronto, ON, Canada); and ^125^I-[Sar^1^, Ile^8^]Ang II (PerkinElmer Japan Co. Ltd., Tokyo, Japan). Three non-peptide AT_1_ receptor ligands (R239470, R781253, and R794847) as candidates of biased ligands were provided by Daiichi Sankyo Co., Ltd., Tokyo, Japan (Figure 1A).

### 2.2. AT_1_ Mutant Receptors

The secondary structure of the AT_1_ receptor is shown in Figure 1B. We mutated four amino acids in transmembrane (TM) III, VI, and VII in the AT_1_ receptor (L112A, Q257A, Y292A, and N295A receptors). We determined the mutation sites of the AT_1_ receptor according to ARB, which produced olmesartan derivatives with different interactions and functions toward the AT_1_ receptor, as previously reported by Zhang et al. [16], since the additional 4-hydroxybenzyl groups of R781253 and R794847 were predicted to extend to a sub-pocket on the bottom of the AT_1_ receptor ligand-binding pocket consisting of Leu^112^ and Gln^257^. In addition, since the biased ligand TRV027 induced changes near the NPxxY motif in TM domain 7 that are related to receptor activation [17], Tyr^292^ and Asn^295^ in the AT_1_ receptor were mutated.

### 2.3. Cell Cultures, Transfection, and Membrane Preparation

The synthetic rat AT_1_ receptor gene, which was cloned in the shuttle expression vector pMT-2, was used for expression and mutagenesis, as described previously [9,15,18]. COS7 cells (No. CRL-1651, ATCC, Manassas, VA, USA) were cultured in 10% fetal bovine serum (FBS) and penicillin- and streptomycin-supplemented Dulbecco’s modified Eagle’s essential medium (PS-DMEM) in 5% CO_2_ at 37 °C. In the experiments, cells were grown without a cell-growth supplement. To express the AT_1_ receptor protein, 10 µg of purified plasmid DNA per 10^7^ cells was used in the transfection. Wild-type and mutated AT_1_ receptors were transiently transfected into cells using Lipofectamine 2000 liposomal reagent (Roche Applied Science, Penzberg, Germany). Cells were plated and grown and were 70–90% confluent at the time of transfection. We incubated plasmid DNA (10 µg)–lipid (10 µL) complexes for 5 min at room temperature and added and incubated the complexes with cells for 24 h. We then analyzed the parameters in transfected cells. Cell viability was >95% according to trypan blue exclusion analysis in the control experiments. Transfected cells that had been cultured for 48–72 h were harvested. Cell membranes were prepared using the nitrogen Parr bomb disruption or freeze–thaw methods.

### 2.4. Competition Binding Study

The *K*_d_ (nM) values of receptor binding were determined with ^125^I-[Sar^1^, Ile^8^]AngII-binding experiments under equilibrium conditions. Binding kinetics were determined following the methods described in the literature [9,15,18]. We carried out binding by using ^125^I-[Sar^1^, Ile^8^] AngII (2200 Ci/mmol) and an increasing concentration (10^−11^–10^−5^ M) of each unlabeled compound. Nonspecific binding of ^125^I-[Sar^1^, Ile^8^] Ang II was determined in the presence of a high concentration (10^−6^ M) of unlabeled [Sar^1^, Ile^8^]. Next, the incubation mixture was separated by vacuum filtration through Whatman GF/C glass filters (Disposable Filter Funnels Grade GF/C, GE Healthcare), and presoaked with 0.1% polyethylenimine. The filter was washed with 3 mL of ice-cold stop buffer (25 mM Tris–HCl, pH 7.4, 1 mM MgCl_2_). The radioactivity trapped on the filters was quantified using an automatic γ-counter. The percentage (%) of specific binding of ^125^I-[Sar^1^, Ile^8^]Ang II was calculated using the following formula: 100 × (% total binding of ^125^I-[Sar^1^, Ile^8^]Ang II—nonspecific binding of ^125^I-[Sar^1^, Ile^8^]Ang II)/(% total binding of ^125^I-[Sar^1^, Ile^8^]Ang II).

### 2.5. IP Production Assay

Ligand (such as agonist or antagonist)-induced IP production through the AT_1_ receptor in transfected COS7 cells transiently expressing the AT_1_ receptor was measured to evaluate cell signaling as a second messenger for controlling blood pressure, since the increases in the intracellular Ca^2+^ concentration were the result of IP accumulation, which indicated vasoconstriction. As a surrogate model with higher levels of AT_1_ receptor expression, COS7 cells are suitable for IP measurement because it is easy to detect differences in IP production. Cells were labeled for 24 h with [^3^H]-myoinositol at 37 °C in DMEM containing 10% FBS. Next, the cells were washed with buffer 3 times and incubated with a medium containing 10 mM LiCl for 20 min; 0.1 µM Ang II or 1 µM ligands were added and incubation was continued for another 30 min at 37 °C. After incubation, the medium was removed, and total soluble IP was extracted from the cells using the perchloric acid extraction method, as described previously [9,15,18]. Cells were plated at a field density of 5 × 10^3^ cells/cm^2^ in 60 mm dishes and cultured in a fresh growth medium at 37 °C in a DMEM 5% CO_2_ incubator. After 24 h of plating, the cells were transfected with empty vector or expression vector for wild-type or mutant AT_1_ receptors. The culture medium was changed 48 h after transfection and 1.5 mL of 1 μCi/mL of [^3^H]-myoinositol was added to each 60-mm dish at 37 °C in DMEM containing 10% FBS. After labeling, the cells were washed with buffer 3 times and incubated with a medium containing 10 mM LiCl for 20 min. The indicated concentration of ligands was added and incubation was continued for an additional 30 min at 37 °C. After incubation, the medium was removed, and total soluble IP was extracted from the cells using the perchloric acid extraction method. Total IP production was expressed as the sum of radioactivity, as described previously [9,15,18].

### 2.6. Immunoblotting of ERK 1/2 Activation

In in vitro studies, ERK 1/2 activation by 1 µM ligands or 0.1 µM Ang II incubated for 10 min was measured using COS7 cells that overexpressed AT_1_ wild-type and mutant receptors. ERK activity was measured in cells maintained for 18 h in DMEM containing 0.1% serum. At the end of the stimulation, the culture medium was removed by aspiration, the cells were washed with ice-cold phosphate-buffered saline, and the dishes were frozen in liquid nitrogen. The cells were scraped using a scraper after melting, and the scraped cells were transferred to a tube. We homogenized the cells in lysis buffer plus protease inhibitors and determined the protein concentration using the bicinchoninic acid method. Immunoblotting was performed with primary antibodies as the specified total ERK (#9102, Cell Signaling, Danvers, MA, USA) or phospho-ERK (#9106, Cell Signaling, Danvers, MA, USA) antibody in each sample [5]. We fractionated total proteins using sodium dodecyl sulfate polyacrylamide gel electrophoresis; we then transferred the proteins to a Hybond nitrocellulose membrane and blocked the blotted membranes with 10% bovine serum albumin in tris-buffered saline buffer (50 mM Tris–HCl pH 7.6, 150 mM NaCl) for 1 h at room temperature. Next, the membranes were incubated with horseradish peroxidase-conjugated secondary antibody (BIO-RAD170-6515 or 170-6516, Bio-Rad Laboratories Inc., Hercules, CA, USA) against total or phosphorylated ERK overnight at 4 °C. An enhanced chemiluminescent substrate system (Amersham, Buckinghamshire, UK) was used. The signal was independently quantified by digital image analysis (ImageJ, National Institutes of Health, Bethesda, MD, USA) [19].

### 2.7. Statistical Analysis

All data are expressed as the mean ± standard error. Each experiment was performed by three or more independent determinations. The significance of differences among the measured values was evaluated with an analysis of variance using an unpaired or paired Student’s *t*-test and Fisher’s *t*-test, as appropriate. Statistical significance was considered to be present at <0.05.

## 3. Results

### 3.1. Binding Affinities of [Sar^1^,Ile^8^]Ang II, Olmesartan, R239470, R781253, and R794847 to AT_1_ Wild-Type and Mutant Receptors

The *K*_d_ values of [Sar^1^,Ile^8^]Ang II, olmesartan, R239470, R781253, and R794847 for the AT_1_ wild-type receptor, as determined by a competition binding study, were 0.8, 2.3, 0.8, 21, and 48 nM, respectively (Table 1). Those of R239470, R781253, and R794847 for the AT_1_ L112A receptor were 37, 23, and 31 nM, respectively. The *K*_d_ values of R239470, R781253, and R794847 were comparable to the *K*_d_ value of olmesartan for the receptor. The *K*_d_ values of R239470, R781253, and R794847 for the AT_1_ Q257 receptor were better than that of olmesartan. Although R239470 could bind to the AT_1_ N295A receptor, R781253 and R794847 could not (*K*_d_ > 10,000 nM).

### 3.2. Levels of IP Production Using Various Ligands in AT_1_-WT, -L112A and Q257A Receptors

The levels of IP production using various ligands in AT_1_ wild-type, -L112A, and Q257A receptors are shown in Figure 2. Ang II significantly increased IP production in AT_1_ wild-type, L112A, and Q257A receptors in this cell system. Olmesartan induced a significant reduction in basal IP production levels in the AT_1_ wild-type, L112A, and Q257A receptors and showed inverse agonism. In the AT_1_ wild-type receptor, R781253, and R794847 significantly decreased and increased IP production, respectively, whereas R239470 did not change IP production (Figure 2). In the AT_1_-L112A and -Q257A receptors, none of the compounds (R239470, R781253, and R794847) affected IP production, although olmesartan induced a significant reduction in basal IP production levels in the receptors.

### 3.3. Levels of ERK Activities Using Various Ligands in AT_1_-WT, -L112A, and Q257A Receptors

Ang II significantly activated ERK1/2 in AT_1_ wild-type, L112A, and Q257A receptors in this cell system (Figure 3). Olmesartan did not change ERK1/2 activation in AT_1_ wild-type, L112A, or Q257A receptors. In the AT_1_ wild-type receptor, R781253 and R794847, but not R239470, activated ERK1/2. R781253 and R794847, but not R239470, activated ERK1/2 in the AT_1_-L112A receptor. None of the compounds activated ERK1/2 in the AT_1_-Q257A receptor.

## 4. Discussion

In this study, we developed a non-peptide AT_1_ receptor compound, R781253, as a biased ligand (Figure 4). The positions of Lys^112^ and Gln^257^ in the AT_1_ receptor may play an important role in IP production with regard to R781253 as a biased ligand, whereas the position of Gln^257^, but not Lys^112^, plays an important role in the ERK1/2 activation of R781253. Thus, a small difference in the molecular structure produced a biased ligand. In addition, the differences in the binding position of the AT_1_ receptor relative to R781253 influenced the function of the biased ligand.

We confirmed that the non-peptide AT_1_ receptor compound R781253 acted as a biased ligand (Figure 4). R239470, R781253, and R794847 were a neutral antagonist, an inverse agonist, and an agonist with regard to IP production, respectively. We previously reported that small differences in the chemical structures of ARBs can be responsible for agonism, neutral antagonism, or inverse agonism regarding IP production toward the AT_1_ receptor [15]. We analyzed the ligand-specific changes in the receptor conformation with respect to stabilization around TM3. Although the agonist-, neutral antagonist and inverse-agonist-binding sites in the AT_1_ receptor are similar, each ligand induced specific conformational changes in TM3. The hydroxyphenyl moiety, which represents the difference between the chemical structures of olmesartan and R794847, interacts with the Leu^112^ of the AT_1_ receptor. Coupled movements of Leu^112^ and Asn^111^ occur in molecular dynamics simulations of the AT_1_ receptor [20]. The flipping of Asn^111^ is essential for Gq-dependent signaling. In this study, we also analyzed the positions of Lys^112^, which is adjacent to Asn^111^, as well as Gln^257^, in the AT_1_ receptor. Leu^112^, in addition to Gln^257^, may play an important role in Gq-dependent IP production with regard to R781253 as a biased ligand. On the other hand, R781253 and R794847, but not R239470, were found to be agonists in terms of ERK1/2 activation. The additional 4-hydroxybenzyl groups of both R781253 and R794847 were predicted to extend to a sub-pocket on the bottom of the AT_1_ receptor ligand-binding pocket, which includes Leu^112^, Lys^199^, Asn^200^, Trp^253^, His^256^, Gln^257^, and Thr^260^ in the AT_1_ receptor [16]. Since the position of Gln^257^, but not of Lys^112^, is important for the ERK1/2 activation by R781253, the hydroxy group in the 4-hydroxybenzyl group of R781253 may be crucial for the binding to Gln^257^ in the AT_1_ receptor for ERK1/2 activation. In addition, the carbamoyl moieties of R239470 and R794847 cannot form a salt bridge to Arg^167^ in the AT_1_ receptor. Therefore, R781253 retains inverse agonism, while R239470 and R794847 do not. In addition, Arg^167^ forms extensive networks of hydrogen bonds and salt bridges with the acidic tetrazole ring and the carboxyl group on the imidazole moiety of olmesartan in the AT_1_ receptor–olmesartan complex. We might have been able to analyze IP production and ERK1/2 activation more effectively using the position of the Arg^167^ mutated AT_1_ receptor. Lys^199^ also played an important role in Ang II binding, as the K199A mutant completely abolishes Ang II binding, and we did not perform any further analysis.

The β-arrestin-biased ligand stabilizes an alternative TM7 conformation in the AT_1_ receptor, whereas G-protein-biased ligands do not favor this conformation [21]. The binding of partial or β-arrestin-biased peptide agonists triggers a shift of TM6, which is a hallmark of GPCR activation, and a change in the conformation of Asn^295^ [20]. In addition, Tyr^292^ and Leu^112^ in the AT_1_ receptor play important roles in coupling Ang II binding to Gq activation. In the active state of AT_1_ receptor structures, the outward rotation of TM6 and its associated inward rotation of TM7 disrupts an inactive state for stabilizing interactions between Asn^111^ and Asn^295^ underneath the ligand-binding pocket. Tyr^292^ is also coupled to the ligand through two adjacent residues, Asn^111^ and Leu^112^, on TM3 [20]. To analyze the importance of the change in TM7 conformation by a non-peptide biased ligand R781253 in this study, we mutated the positions of Tyr^292^ (Y292A) and Asn^295^ (N295A) in TM7. Since the *K*_d_ values of R781253 against Y292A and N295A were 95 nM and >10,000 nM, respectively, and since the binding affinities were very low, we did not confirm IP production or ERK activation using R781253 in the AT_1_-Y292A and -N295A receptors. Further study is needed to resolve these issues.

Activation of the Gq/11 pathway is associated with a specific conformational transition that is stabilized by the agonist. On the other hand, activation of the β-arrestin pathway is linked to the stabilization of the ground state of the receptor [22]. Distinct conformations of the AT_1_ receptor are associated with distinct cell signaling pathways [23]. The activities of GPCRs are finely regulated by GPCR kinases (GRKs), which phosphorylate agonist-occupied receptors and start the process of desensitization. GRK2 and GRK5 are predominantly expressed in the heart, where they have both canonical and non-canonical functions. Biased ligands induce the recruitment of distinct subtypes of GRK to the receptor as compared with balanced ligands. β-arrestin induces sustained G protein-induced cell signaling by a ternary GPCR–G-protein–β-arrestin complex. The activation of the G protein triggers the initiation of the β-arrestin system by releasing free Gβγ subunits and activating GRK2/3. Moreover, while β-arrestin recruitment depends on both GRK5/6 and GRK2/3 upon the binding of Ang II, it depends solely on GRK5/6 upon the binding of the β-arrestin-biased ligand TRV027 [24]. In this study, although we did not analyze the conformational transition or the importance of the kinds of GRK of R239470, R781253, and R794847, the biased ligand R781253 may induce a distinct conformational transition and the recruitment of different GRKs.

AT_1_ receptor signaling has been implicated in Severe Acute Respiratory Syndrome Coronavirus-2 (SARS-CoV-2) infection [25]. SARS-CoV-2 enters cells by binding to angiotensin converting enzyme 2 (ACE2), which lead to the downregulation of ACE2 activity. Ang II, the ligand for the AT_1_ receptor, activates a Gq signaling pathway, leading to vasoconstriction and adverse cellular responses in heart failure and acute lung injury. On the other hand, the action of Ang II is moderated by β-arrestin [25]. Cleavage of Ang II by ACE2 decreases Ang II levels and increases Ang(1-7) production, which is an endogenous β-arrestin-biased agonist of the AT_1_ receptor with potential therapeutically beneficial effects. Compared with ARBs, the biased agonist of AT_1_ receptor R781253, which can improve potency and pharmacodynamic properties compared with Ang(1-7), may be more advantageous in patients with SARS-CoV-2 infection because R781253 blocks Gq-dependent Ang II activity and induces β-arrestin signaling.

A limitation of this study is that it was conducted under limited conditions and in vitro, and no research has been conducted in vivo within the human body. However, it is considered to have sufficient reliability for verifying the binding mode between the ligand and receptor and subsequent specific intracellular signaling pathways. In addition, based on previous reports, mutated receptors were created and verified. Nonetheless, we performed experiments with only one type of cell. Each type of cell has different intracellular signal transduction systems, and the strength of the signals varies, so the results obtained in this study do not necessarily occur universally in all kinds of cells.

## 5. Conclusions

R239470, R781253, and R794847 were shown to be a neutral antagonist, an inverse agonist, and an agonist with regard to IP production, respectively. On the other hand, R781253 and R794847, but not R239470, were agonists with respect to ERK1/2 activation. Thus, we developed a non-peptide AT_1_ receptor compound as a biased ligand.

## Figures and Tables

**Figure 1 biomedicines-11-01486-f001:**
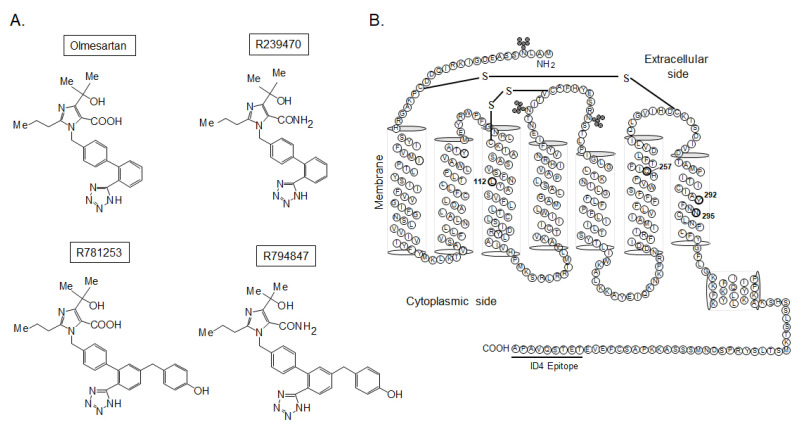
(**A**) Construction of olmesartan and its related compounds. (**B**) Secondary structure of the angiotensin II type 1 (AT_1_) receptor.

**Figure 2 biomedicines-11-01486-f002:**
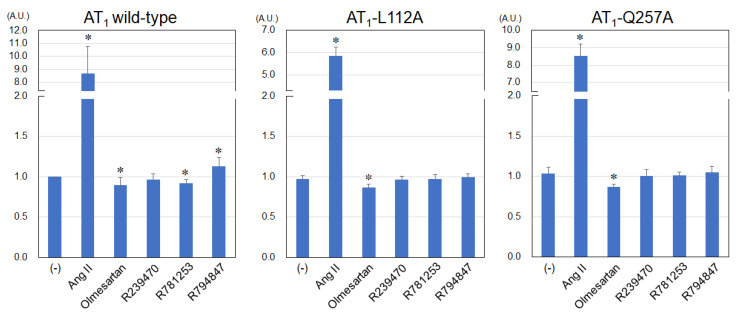
Levels of IP production using various ligands in AT_1_ wild-type, -L112A, and Q257A receptors. IP, inositol phosphate; AT_1_, angiotensin II type 1; A.U. arbitrary unit. n = 5. * *p* < 0.05 vs. (-).

**Figure 3 biomedicines-11-01486-f003:**
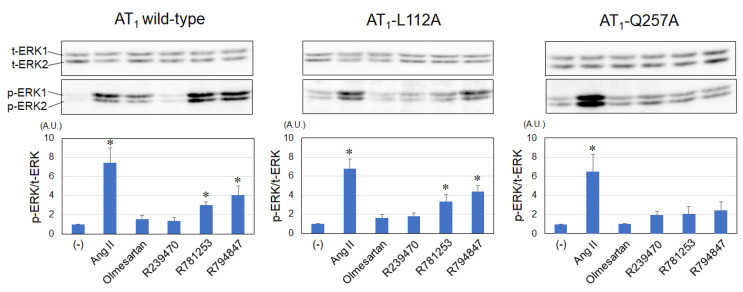
Levels of ERK activities using various ligands in AT_1_ wild-type, -L112A, and Q257A receptors. t-ERK, total extracellular signal-regulated kinase; *p*-ERK, phospho-ERK; AT_1_, angiotensin II type 1; A.U. arbitrary unit. n = 5. * *p* < 0.05 vs. (-).

**Figure 4 biomedicines-11-01486-f004:**
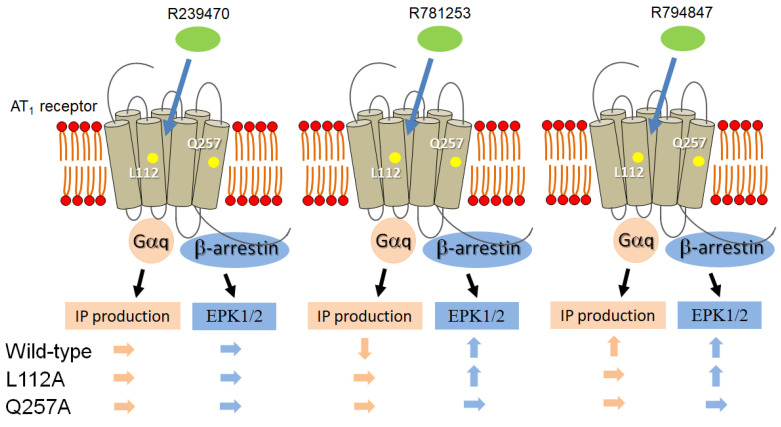
Summary of IP production and ERK activities by olmesartan and its related compounds in AT_1_ wild-type, -L112A, and Q257A receptors. IP, inositol phosphate; ERK, extracellular signal-regulated kinase.

**Table 1 biomedicines-11-01486-t001:** Binding affinities (*K*_d_, nM) of [Sar^1^,Ile^8^]Ang II, olmesartan, R239470, R781253, and R794847 to AT_1_ wild-type and mutant receptors (n = 3).

Receptors	Wild-Type	L112A	Q257A	Y292A	N295A
Ligands					
[Sar^1^,Ile^8^]Ang II	0.8 ± 0.4	1.3 ± 0.3	3.3 ± 1.5	2.1 ± 0.9	7.0 ± 2.0
Olmesartan	2.3 ± 0.8	27 ± 7	228 ± 38	47 ± 15	334 ± 105
R239470	0.8 ± 0.3	37 ± 12	20 ± 2	6.3 ± 3.8	69 ± 8
R781253	21 ± 10	23 ± 7	14 ± 6	95 ± 7	>10,000
R794847	48 ± 12	31 ± 5	2.7 ± 0.8	30 ± 8	>10,000

## Data Availability

The data presented in this study are available are available from the corresponding author on reasonable request.

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
