# Peer review of "Development of a Non-Peptide Angiotensin II Type 1 Receptor Ligand by Structural Modification of Olmesartan as a Biased Agonist"

_biomedicines, 2023, doi:10.3390/biomedicines11051486_

Round 1

Reviewer 1 Report

In the present study authors report synthesis of 3 novel AT1 receptor ligands and their activity regarding increase in intracellular inositol phosphate concentration and stimulation of extracellular signal-regulated kinases. The topic and the results are of interest and the manuscript is well-written. However, there are also some concerns to be addressed.

1)     Section 2.6: the style of the method description should be revised. In the present form, the text looks like the kit instruction. It should be described what WAS done by the authors; the past form should be used consistently.

2)     Line 168: the software used for densitometric analysis should be specified.

3)     It is not entirely clear why these mutated receptors were used. It is important how new ligands affect wild-type mediated signaling but the implications of activity through mutant receptors are not clear.

4)     The significant limitation of this study was that only one cell type was used. This issue should be discussed.

Author Response

Response to Reviewer #1

Ref. : Ms. No. biomedicines-2178737R1
Development of non-peptide angiotensin II type 1 receptor ligand by structural modifications of olmesartan as a biased agonist

Authors: Matsuo Y, et al.

We thank this editor very much for reviewing our manuscript. We appreciate the Reviewer's kind suggestions and have revised our manuscript according to the Reviewer's comments.

Point-by-point response to the editor's comments:

Section 2.6: the style of the method description should be revised. In the present form, the text looks like the kit instruction. It should be described what WAS done by the authors; the past form should be used consistently.

Ans.) Thank you for your comments. We revised them according to your comments.

Line 168: the software used for densitometric analysis should be specified.

Ans.) Thank you for your comment. We revised it according to your comment.

It is not entirely clear why these mutated receptors were used. It is important how new ligands affect wild-type mediated signaling but the implications of activity through mutant receptors are not clear.

Ans.) Thank you for your suggestion. We know from another previous papers and from our paper about the binding sites of olmesartan to the AT1 receptor. The candidates of ligands created at this time were synthesized by mimicking olmesartan, and it is thought that the binding sites of the biphenyl-tetrazole group, which is the main skeletal stracture of olmesartan, to the AT1 receptor does not change.

The significant limitation of this study was that only one cell type was used. This issue should be discussed.

Ans.) Thank you for your comment. We agree with you and added the sentences to the study limitation.

Reviewer 2 Report

The authors presented interesting findings about developing non-peptide angiotensin II type 1 receptor ligands based on Olmesartan. 

Major suggestions:
1. Rows 180-182: As seen from Table 1 (rows 189-193, column 3 L112A), the Kd of R239470  (37 nM) is NOT 10-fold higher than olmesartan (27 nM) for AT1 L112A receptor. Please correct this adequately.

2. Rows 195-203: Text can be removed since the same text for Figure 2 is written in Rows 208-216.

3. Please correct Figure 3, indicating properly ERK1 and ERK2 in the immunoblots for t-ERK and p-ERK. As visible from the AT1- wild-type immunoblot, Olmesartan increases p-ERK1/2 following treatment.

4. It is also not proven that R781253 cause a significant phosphorylation in AT1-L112A (Figure 3). Additional proof is needed for acceptance of this statement since the immunoblot technique is a semi-quantitate

5. Please, describe in detail the antibodies used for t-ERK and p-ERK analysis. The antibodies information is missing and not defined in citation 5, nor the reference 12 cited within reference 5.

6. To prove the observations, please submit at least 3 immunoblots in supplementary materials for t-ERK and p-ERK in AT1 wild-type, AT1-L112A, and AT1-Q257A.

7. In the title, please indicate that the substances studied are modifications of Olmesartan.

8. Please add DOI numbers in all references.

Minor:

Row 20: L112A plays a key role in IP production – to be removed or placed upper

Row 114: harvested, Cell – Should be the period at the end of the sentence

Row 119-120: were determined following methods described in [9, 15, 18]

Row 126: The radioactivity by trapped – remove “by”

Row 143: 5 x 103 cells/cm2 in 60-mm dishes – Should be the power of 3.

Rows 144-145: Аt 24 h after plating, the cells were transfected with….

Row 168: Please describe the digital image analysis in the Material and Methods.

In figures’ legends indicate the number of experiments used for statistical analysis.

Please, remove the repeated sentences and improve the English writing and grammar.

Author Response

Response to Reviewer #2

Ref. : Ms. No. biomedicines-2178737R1
Development of non-peptide angiotensin II type 1 receptor ligand by structural modifications of olmesartan as a biased agonist

Authors: Matsuo Y, et al.

We thank this editor very much for reviewing our manuscript. We appreciate the Reviewer's kind suggestions and have revised our manuscript according to the Reviewer's comments.

Point-by-point response to the reviewer's comment:

Major suggestions:

  1. Rows 180-182: As seen from Table 1 (rows 189-193, column 3 L112A), the Kd of R239470  (37 nM) is NOT 10-fold higher than olmesartan (27 nM) for AT1 L112A receptor. Please correct this adequately.

Ans.) Thank you for your comment. We corrected it.

  1. Rows 195-203: Text can be removed since the same text for Figure 2 is written in Rows 208-216.

Ans.) Thank you for your comment. We removed them.

  1. Please correct Figure 3, indicating properly ERK1 and ERK2 in the immunoblots for t-ERK and p-ERK. As visible from the AT1- wild-type immunoblot, Olmesartan increases p-ERK1/2 following treatment.

Ans.) Thank you for your comment. We removed them.

  1. It is also not proven that R781253 cause a significant phosphorylation in AT1-L112A (Figure 3). Additional proof is needed for acceptance of this statement since the immunoblot technique is a semi-quantitate.

Ans.) We showed 3 immunoblots in Supplementary Figure and there were some statistical significant activation in Figure 3.

  1. Please, describe in detail the antibodies used for t-ERK and p-ERK analysis. The antibodies information is missing and not defined in citation 5, nor the reference12 cited within reference 5.

Ans.) Thank you for your suggestion. We added the information.

  1. To prove the observations, please submit at least 3 immunoblots in supplementary materials for t-ERK and p-ERK in AT1 wild-type, AT1-L112A, and AT1-Q257A.

Ans.) We showed them in Supplementary Figure.

  1. In the title, please indicate that the substances studied are modifications of Olmesartan.

Ans.) Thank you for your suggestion. We changed the title “Development of non-peptide angiotensin II type 1 receptor ligand by structural modifications of olmesartan as a biased agonist”.

  1. Please add DOI numbers in all references.

Ans.) Thank you for your comments. We revised them.

Minor:

Row 20: L112A plays a key role in IP production – to be removed or placed upper

Ans.) We removed the sentence.

Row 114: harvested, Cell – Should be the period at the end of the sentence

Ans.) We corrected it.

Row 119-120: were determined following methods described in [9, 15, 18]

Ans.) We corrected the sentence.

Row 126: The radioactivity by trapped – remove “by”

Ans.) We corrected it.

Row 143: 5 x 103 cells/cm2 in 60-mm dishes – Should be the power of 3.

Ans.) We corrected it.

Rows 144-145: Аt 24 h after plating, the cells were transfected with….

Ans.) We corrected it.

Row 168: Please describe the digital image analysis in the Material and Methods.

In figures’ legends indicate the number of experiments used for statistical analysis.

Ans.) We described “The results are expressed as the mean ± standard error of three or more independent determinations” in the section of statistical analysis.

Round 2

Reviewer 1 Report

The manuscript has been revised according to the reviewers' comments. All concerns raised by the reviewers have been adequately addressed by the authors.

Author Response

Response to Reviewer #1

Ref. : Ms. No. biomedicines-2178737R2
Development of a non-peptide angiotensin II type 1 receptor ligand by structural modification of olmesartan as a biased agonist

Authors: Matsuo Y, et al.

We thank this editor very much for reviewing our manuscript.

Reviewer 2 Report

Thank you very much to the authors for improving the manuscript, but the observations need to be proved by providing the correct immunoblots!

As visible from the immunoblots in the Supplementary Figure:

 1.       p-ERK (AT1-L112A)- 1st experiment (blot) is same as 2nd experiment and same as 3rd experiment with low film exposure time. Please, provide 3 results (blots) from independent experiments.

2.      p-ERK (AT1-Q257A)- 2nd experiment has 7 lines, while t-ERK has 6. Please, show the appropriate one.

Please describe the number of experiments used for statistical analysis in figures and tables' legends. Since in "2.7. Statistical analysis" is written, "…mean ± standard error of three or more …. "

There is a need for moderate editing of the English language. 

Author Response

Response to Reviewer #2

Ref. : Ms. No. biomedicines-2178737R2
Development of a non-peptide angiotensin II type 1 receptor ligand by structural modification of olmesartan as a biased agonist

Authors: Matsuo Y, et al.

We thank this reviewer very much for reviewing our manuscript. We appreciate the Reviewer's kind suggestions and have revised our manuscript according to the Reviewer's comments.

Point-by-point response to the reviewer's comment:

1. p-ERK (AT1-L112A)- 1st experiment (blot) is same as 2nd experiment and same as 3rd experiment with low film exposure time. Please, provide 3 results (blots) from independent experiments.

Ans.) Thank you for your comment. We are sorry, and we have corrected the mistakes.

2. p-ERK (AT1-Q257A)- 2ndexperiment has 7 lines, while t-ERK has 6. Please, show the appropriate one.

Ans.) Thank you for your comment. We are sorry, and we have corrected the mistakes.

Please describe the number of experiments used for statistical analysis in figures and tables' legends. Since in "2.7. Statistical analysis" is written, "…mean ± standard error of three or more …. "

Ans.) Thank you for your comment. We added the number.

Round 3

Reviewer 2 Report

The authors presented interesting findings about developing non-peptide angiotensin II type 1 receptor ligands based on Olmesartan. The manuscript can be considered for publication.

The English language can be improved.